# Continual Learning via Explicit Structure Learning

## Abstract

Despite recent advances in deep learning, neural networks suffer catastrophic forgetting when tasks are learned sequentially. We propose a conceptually simple and general framework for continual learning, where structure optimization is considered explicitly during learning. We implement this idea by separating the structure and parameter learning. During structure learning, the model optimizes for the best structure for the current task. The model learns when to reuse or modify structure from previous tasks, or create new ones when necessary. The model parameters are then estimated with the optimal structure. Empirically, we found that our approach leads to sensible structures when learning multiple tasks continuously. Additionally, catastrophic forgetting is also largely alleviated from explicit learning of structures. Our method also outperforms all other baselines on the permuted MNIST and split CIFAR datasets in continual learning setting.

## 1 Introduction

Learning different tasks continuously is a common and practical scenario that happens all through the course of human learning. The learning of new skills from new tasks usually does not have negative impact on the previously learned tasks. Furthermore, with learning multiple tasks that are highly related, it often helps to advance all related skills. However, this is commonly not the case in current deep learning models. When presented a sequence of learning tasks, the model experiences so called "catastrophic forgetting" problem (McCloskey & Cohen, 1989; Ratcliff, 1990), where the model "forgets" the previous learned task while learning the new task. This is an interesting phenomenon that has attracted lots of research recently.

Many efforts have been tried to overcome catastrophic forgetting. Common approaches such as Elastic weight consolidation (EWC Kirkpatrick et al., 2017) and synaptic intelligence (Zenke et al., 2017) focuses on alleviating catastrophic forgetting by applying constraints on the update of the parameters. However, forgetting is still non-negligible with these approaches, especially when the number of tasks increases. Forgetting could also be alleviated with memory-based methods, where certain information regarding learned tasks are stored to help retaining the performance of the learned tasks (see Lopez-Paz et al., 2017; Sener & Savarese, 2018, for example). Additionally, there are also methods (Mallya & Lazebnik, 2018; Rebuffi et al., 2017a; 2018; Mancini et al., 2018) that could learn multiple domains and completely avoid forgetting by adding a small portion of parameters while the original weights is kept fixed. However, these kind of models rely on a strong base network and knowledge can only be transferred from one task to another.

Most of the current approaches in continual learning employ a model with a fixed structure, and apply the same model to learn all tasks. We believe that a more intuitive and sensible approach is to learn task specific model structure *explicitly* along with learning of the model parameters[1]. Since each task may require different architecture. In case two tasks are not quite relevant, it may not make much sense to employ the same structure for learning. For example, in case of learning digit and face recognition models, the lower level features required is likely to be drastically different, and similarly for the required structure. Forcing the same structure for these tasks is likely to prevent

---

[1]The structure that referred here is more fine-grained, such as number of layers, type of operations at each layer, etc. It does not refer to generic structure names like convolutional nerual networks or recurrent neural networks.

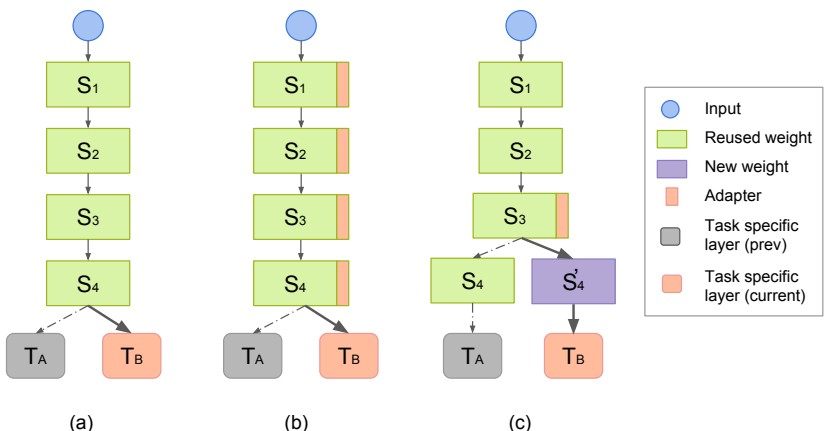

Figure 1: Illustration of different approaches for continuous learning. a) All but the task specific layer are shared, catastrophic forgetting is countered by techniques that prevents parameters to move to lower energy regions of previous tasks. b) Each task will add some fixed task specific parameters, all layers' original weights are not tuned, and thus prevents forgetting. c) Our approach, where network structure is determined by architecture search. In this example, the search results decide to "reuse" the first two layer, do "adaptation" for the third layer and allocate "new" weight for the fourth layer.

sharing, since very less information is relevant between the tasks. On the other hand, if tasks were allowed to use different structure, it has the potential to encourage information sharing. This is because now the irrelevant part can be handled by the new/different structure.

Therefore, in this work we propose a framework that explicitly separate the learning of model structure and model parameters. In particular, we employ architecture search for each of the sequential tasks to find the optimal structure for the current task. The search considers various options, such as share previous layer's parameter, spawn new parameters, and so on. The model parameters are then learned correspondingly. We found that 1) qualitatively, the learned structure is sensible – similar tasks tends to share more parameters and structure, whereas distant tasks share less; 2) quantitatively, separating the structure and parameter learning significantly reduced catastrophic forgetting. Additionally, we propose a new approach for doing continuous learning with architecture search.

## 2 Explicit Structure Learning

The continual learning task that we are concerned with in this work is defined as follows. $N$ tasks $T_1, T_2, \ldots, T_N$ forms a task sequence $\mathbf{T} = (T_1, T_2, ..., T_N)$. Each task $T_i$ is composed of dataset $\mathcal{D}_i = \{(x_1^{(i)}, y_1^{(i)}), (x_2^{(i)}, y_2^{(i)}), \ldots, (x_{N_{T_i}}^{(i)}, y_{N_{T_i}}^{(i)})\}$, where each dataset is of size $N_{T_i}$. The model gets to observe tasks from 1 to $N$ sequentially. After the model finished learning on task $T_i$, the model can no longer access this task, i.e. all data from $\mathcal{D}_i$ will not be available when learning tasks $T_{i+1}$ to $T_N$.

Ideally, we would like to minimize the following loss function in this continuous learning setting

$$\mathcal{L}(\theta) = \sum_{i=1}^{N} \mathcal{L}_i(\theta) \tag{1}$$

$$\mathcal{L}_i(\theta) = \frac{1}{N_{T_i}} \sum_{n=1}^{N_{T_i}} \ell_i(f_\theta(x_n^{(i)}), y_n^{(i)}) \tag{2}$$

where $f_\theta$ is the model and $\ell_i$ is the loss function for task $T_i$. However, since we do not have access to all dataset at the same time, the above loss in equation 1 is very hard to minimize. In case we ignore the access issue and train each task separately, this will result in catastrophic forgetting. Additionally, the above loss definition clearly does not account for structure changes explicitly in the

formulation. As motivated earlier, we believe explicit learning of structure is important for continuous learning. The key intuition for explicit learning structures for new tasks is that in case there is high dissimilarity between the current and previously seen tasks, optimizing model parameter may inevitably cause forgetting. This is because the model is asked to function very differently, and it is very unlikely that the new optimal parameters will also be a good solution for the previous tasks. Even in case where the tasks are similar, it may not be ideal to share all of the structure, as there might be fine grained details in tasks that make the part of the model focus on extracting different types of representations. Therefore, we introduce here $s_i(\theta)$ to indicate the structure for task $T_i$. The updated loss for individual task is then

$$\mathcal{L}_i(\theta) = \frac{1}{N_{T_i}} \sum_{n=1}^{N_{T_i}} \ell_i(f_{s_i(\theta)}(x_n^{(i)}), y_n^{(i)}) \tag{3}$$

Now the structure is explicitly taken into consideration when learning all the tasks. When optimizing the updated loss in equation 3, one needs to determine the optimal parameter based on the structure $s_i$. This loss can be viewed in two ways. One can interpret the above loss as selecting a task specific network from a 'super network' that has parameter $\theta$ using $s_i$, or for each task we train a new model with parameter $s_i(\theta)$. There is a subtle difference between this two views. The former one has an constraint on the total model size, where as the latter one does not. So in worst case scenario, the model size will grow linearly as we increase the number of tasks. This would lead to a trivial solution – training completely different models for different tasks and is no longer continuous learning! To address this problem, we propose the following loss

$$\mathcal{L}_i(\theta) = \frac{1}{N_{T_i}} \sum_{n=1}^{N_{T_i}} \ell_i(f_{s_i(\theta)}(x_n^{(i)}), y_n^{(i)}) + \beta_i \mathcal{R}_i^s(s_i) + \lambda_i \mathcal{R}_i^p(\theta) \tag{4}$$

where $\beta_i > 0$, $\lambda_i \geq 0$, $\mathcal{R}_i^s$ and $\mathcal{R}_i^p$ indicate regularizer for structure and parameter, respectively. For instance, one can use $\ell_2$ regularization for $\mathcal{R}_i^p$ when optimizing model parameters, and $\mathcal{R}_i^s$ can be as simple as the (log) number of parameters. In this way, the total number of parameters are upper bounded, and the degenerate case is thus avoided.

## 3 OUR IMPLEMENTATION

It is a challenging problem to optimize the loss described in equation 4, since it involves explicit optimization of the structure of the model. In our implementation, we choose to separate this optimization problem to two steps: structure optimization and model parameter learning. In particular, we employ neural architecture search to deal with structure optimization, and then gradient based method for learning model parameters after the structure is fixed. We explain our implementation in detail in the following section.

### 3.1 STRUCTURE OPTIMIZATION

We employ neural architecture search for structure optimization. Before we move on to further details, we make a further simplification, where we assume that one already have in mind a global structure that may work for all tasks, and we are only selecting connectivity pattern between layers and their corresponding operator. It is straight forward to adapt this to more complicated cases, we make the simplification because: 1) it is common in a multi-task continual learning scenario that one has some rough clue regarding the overall model structure; 2) this simplifies the optimization problem significantly.

Let's define a certain network with $L$ shareable layers and one task-specific layer (i.e. last layer) for each task. A super network $\mathcal{S}$ is maintained so that all the new task-specific layers and new shareable layers will be stored into $\mathcal{S}$.

The goal of search is trying to find out the optimal choice for each of the $L$ layers, given the current task data $\mathcal{D}_i$ and all the shareable layer's weights stored in $\mathcal{S}$. The candidate choices for each layer could be "reuse", "adaptation" and "new". The reuse choice will make new task use the same parameter as the previous task. The adaptation option adds a small parameter overhead that trains an additive function to the original layer output. The new operator will spawn new parameters of

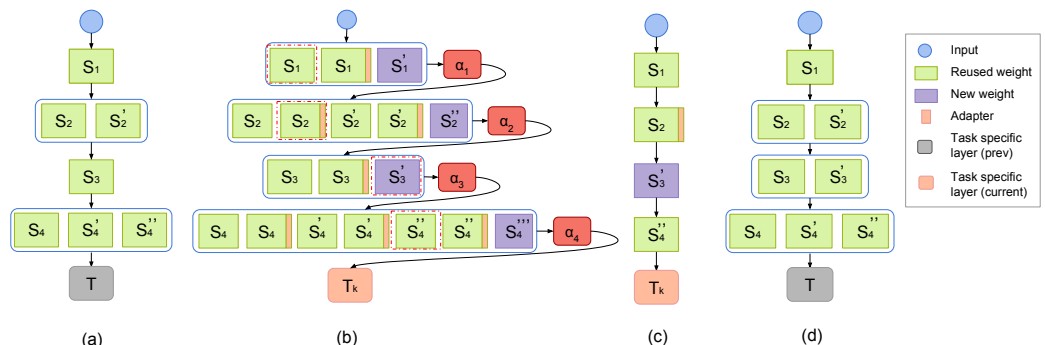

Figure 2: Illustration of the training pipeline of our framework. a) Current state of super model. In this example, the 1st and 3rd layers have single copy of weight, while the 2nd and 4th has two and three respectively. b) During search, each copy of weight for each layer will have a "reuse" and an "adaptation" options plus a "new" option, thus totally $2|\mathcal{S}^l| + 1$ choices. $\alpha$ is the weight parameters for the architecture. c) Parameter optimization with selected architecture on the current task k. d) Update super model to add the newly created $S_3'$.

exactly the size of the current layer parameters. Here, we denote the size of the $l_{th}$ layer in super network $\mathcal{S}$ as $|\mathcal{S}^l|$. The total number of choices in the $l_{th}$ layer $C_l$ is $2|\mathcal{S}^l| + 1$, because we will have $|\mathcal{S}^l|$ "reuse", $|\mathcal{S}^l|$ "adaptation" and 1 "new". Thus, the total search space is $\prod_l^L C_l$. One potential issue here is that, in worst case, the search space may grow exponentially with respect to the number of tasks. One way of dealing with this is to limit the total number of possible choices, and make use a priority queue for managing the options. However, we do not find this necessary in all of our experiments.

Similar to DARTS (Liu et al., 2018), to make the search space continuous, we relax the categorical choice of the $l_{th}$ layer as a softmax over all possible $C_l$ choices:

$$x_{l+1} = \sum_{c=1}^{C_l} \frac{\exp(\alpha_c^l)}{\sum_{c'=1}^{C_l} \exp(\alpha_{c'}^l)} g_c^l(x_l) \tag{5}$$

Here, the vector $\alpha^l$ of dimension $C_l$ is the architecture weights that are used for mixing the choices for each sharable layer. And $g_c^l$ here is the operator for the choice $c$ at layer $l$ which is expressed as:

$$g_c^l(x_l) = \begin{cases} S_c^l(x_l) & \text{if } c \leq |\mathcal{S}^l|, \\ S_c^l(x_l) + \gamma_{c-|\mathcal{S}^l|}^l(x_l) & \text{if } |\mathcal{S}^l| < c \leq 2|\mathcal{S}^l|, \\ o^l(x_l) & \text{if } c = 2|\mathcal{S}^l| + 1 \end{cases} \tag{6}$$

Here, $\gamma$ is the adapter operator and $o$ is the new operator training from scratch. After this relaxation, the task of discrete search become optimizing a set of continuous weights $\alpha = \{\alpha^l\}$. After finishing the searching, the optimal architecture could be obtained by taking the index with the largest weight $\alpha_c^l$ for each layer $l$, i.e. $c_l = \arg\max \alpha^l$.

Adopting the training strategy from DARTS. We use validation loss $L_{val}$ to update the architecture weights $\alpha$, while the operator weights are optimized by the training loss $L_{train}$. The architecture weights and operator weights are updated alternately during the search process. Because it is a nested bi-level optimization problem, the original DARTS provide a second-order approximation for more accurate optimization. In this work, we find it is sufficient to use the simple alternately update approach, which was referred as the first-order approximation in (Liu et al., 2018).

To make it clear how "reuse", "adaptation" and "new" operator works, we walk through a concrete example in the following. Let us take a convolutional neural network (CNN) with all the layers using 3x3 kernel size as an example. The choice of "reuse" is just using the existing weight and keep it fixed during learning, thus there is no additional parameter cost. For "adaptation", it could be a 1x1 conv layer added to the original 3x3 conv layer in parallel (i.e. similar to the adaptor used in paper (Rebuffi et al., 2017a)). During training, the weight of the original 3x3 conv is kept fixed, while the parameters of the 1x1 conv adapter is modified. In this case, the additional parameter cost is

only 1/9 of the original parameter size. In case of "new", it is literally start with a new operator that initialized randomly and train from scratch. We make use of the loss function $L_{val}$ to implement the regularizer $\mathcal{R}_i^s(s_i)$. The value of the regularizer is set proportional to the product of the additional parameter size $z_c^l$ and its corresponding weight $\alpha_c^l$ (i.e. $\mathcal{R}_i^s(s_i) = \sum_{c,l} \alpha_c^l z_c^l$). The architecture weights $\alpha$ is optimized in terms of both accuracy and parameter efficiency at the same time.

## 3.2 PARAMETER OPTIMIZATION

After we get the optimal choices for each layer from the search procedure, we retrain the optimal architecture on the current task. There are two strategies to deal with "reuse", we can either fix it unchanged during retraining just as in search, or we can tune it with some regularization – simple $\ell_2$ regularization or more sophisticated regularizations like elastic weight consolidation (Kirkpatrick et al., 2017). The former strategy could avoid forgetting completely, however it will lose the chance of getting positive backward transfer, which means the learning of new tasks may help previous tasks' performance. When the search process select "reuse" at layer $l$, it means that the $l_{th}$ layer tends to learn very similar representation as it learned from one of the previous tasks. This is an indication of semantic similarity learned at this layer $l$ between the two tasks. Thus, we conjecture that tuning the $l_{th}$ layer with some regularization could also benefit the previous tasks, or at least reduce catastrophic forgetting due to the semantic relationships. In the experiment section, we investigate this hypothesis in more detail. Finally, after retrained on the current task, we need to update/add the created and tuned layers, task-specific adapters and classifiers in the maintained super network. It will be used for model inference and also can be the basis for future architecture search on new tasks.

## 4 HOW EXPLICIT STRUCTURE LEARNING AFFECTS CONTINUOUS LEARNING

In this section, we test two main hypothesis that leads to our framework. First, does making structure learning explicit leads to discovery of sensible model architectures for corresponding tasks? Second, if so, does this leads to better continuous learning? We test these two hypothesis on two datasets: permuted MNIST and the visual domain decathlon dataset (Rebuffi et al., 2017a). The permuted MNIST dataset is a simple image classification problem that derived from the MNIST handwritten digit dataset (Yann LeCun, 1998), which is commonly used as benchmark in continual learning literature (Kirkpatrick et al., 2017; Lopez-Paz et al., 2017; Zenke et al., 2017). For each task, a unique fixed random permutation is used to shuffle the pixels of each image, but the label corresponding to the image is kept fixed. The visual decathlon dataset consists of 10 image classification tasks – ImageNet, CIFAR100, Aircraft, DPed, Textures, GTSRB, Omniglot, SVHN, UCF101 and VGG-Flowers. The images of all the tasks are resized to lower-edge be 72 pixels. And the tasks are across multiple domains and dataset sizes are highly imbalanced, which makes it a good candidate to investigate the continual learning problem with potential inter-task transfer.

For all MNIST experiments, we use a 4-layer fully-connected neural networks with 3 feed-forward layers and the $4th$ layer is the *shared* softmax classification layer accross all tasks, as shown in Figure 3(a). This corresponds to the so called 'single head' setup (Farquhar & Gal, 2018). We choose to use this setting because for permuted MNIST dataset the semantic of all tasks are all the same, and we think sharing is a more reasonable design choice. We apply our implementation of the framework for learning 10 permuted MNIST tasks in sequence. For simplicity, we only use two choices, "reuse" and "new", during the structure optimization.

For experiments on visual decathlon dataset, we use a 26-layer ResNet (He et al., 2016) as the base network to learn the first task. This network consists of 3 residual blocks, each output 64, 128, 256 channels. Each residual block contains 4 residual units, each of which consists of two convolutional layers with $3 \times 3$ kernels and a skip connection. At the end of each residual block, the feature resolution is halved by average pooling. We adopt three type of choices during the search, i.e. "reuse", "adaptation" and "new". For "adaptation", a $1 \times 1$ convolution layer with channels equal to the current layer output channels is added for the corresponding layer. The convolved results are then added back to the convolution output. During training, the weight of the original layer is kept fixed, while the $1 \times 1$ layer is adjusted.

### 4.1 DOES EXPLICIT STRUCTURE LEARNING LEADS TO SENSIBLE STRUCTURES?

We first experimented on the permuted MNIST dataset. One would expect a sensible architecture for these tasks tends to share higher level structures but differs at lower layers. This is because the high level semantic information of digits is identical throughout, which should imply similar high level structures. However, since the input pixels are all shuffled differently, the lower level features would need to be different. This implies different parameter values should be used in lower layers. Interestingly, our experiments show that the structure optimization chooses the same configuration that make brand new first layer and reuse all of the following layers in all the 10 tasks (see Figure 3 b). This exactly confirms the prediction that we had before, which is a good indication that the structure optimization is working properly.

To further demonstrate that the learned structure indeed make sense, we performed additional experiments. Since during the search process on each new task, our model always trying to "new" the first layer and "reuse" the other layers. So, the simple question could be raised here is that what if we use new parameters for other layers? Because the feed-forward layers has similar input and output dimensions, it's fair to compare the five setups that the $ith$ layer is "new" and others are "reuse". In the results, we found that using new parameters in the first layer is actually the best choice compared with other situations (see Fig. 3 d). Making the second layer as "new" also results in less forgetting compared with making one of the last three layers as "new".

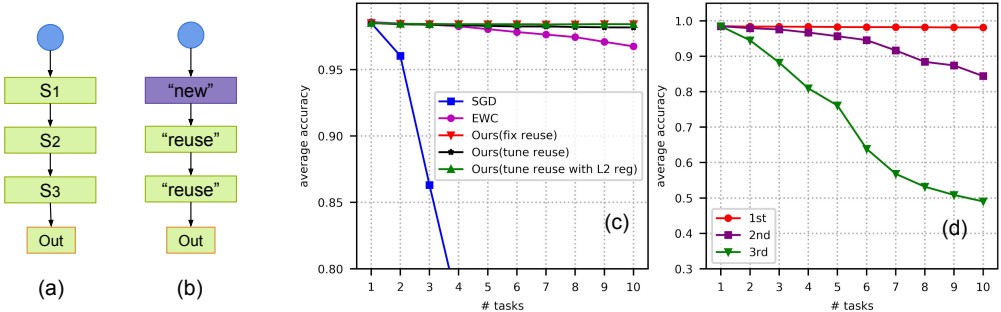

Figure 3: Results on permutated MNIST dataset. a) The base network structure with 3 fully-connected layers and 1 output layer. b) The searched architecture results from running 10 permutated MNIST sequentially. The first layer always choose to "new", while other layers choose to "reuse". c) Comparing our method (fix, tune reuse with and without regularization) with SGD and EWC on the average accuracy over the seen tasks. d) Ablation experiments of "new" different layers in terms of average accuracy over the seen tasks.

Next, we run experiments on the visual decathlon dataset, which is a more realistic image dataset. To see if the results from structure optimization is consistent with what we have previously, we did two experiments. We first test on a similar task pair, i.e. ImageNet and CIFAR-100. In this case, the images are all natural images, although the size of objects are largely different, most of the representation should still be very similar. Indeed, as we can see from left of Fig. 5, most of the layers are shared for these two tasks. Next, we tested on two drastically different tasks, i.e. ImageNet and Omniglot. Since these are two very different visual tasks, we would expect minimal sharing in the resulting structure. As can be see from right of Fig. 5, most of the resulting structure leads to use option "new".

The above experiments suggest that structure optimization does result in sensible task specific structures. The learned task structure tends to share when the semantic representation of corresponding layers are similar, and spawn new parameters when the required information is very different.

### 4.2 DOES EXPLICIT STRUCTURE LEARNING AFFECTS FORGETTING?

With first hypothesis confirmed, the next natural question is – do sensible structures lead to better continuous learning capability. More specifically, will explicit structure learning help catastrophic forgetting in any way?

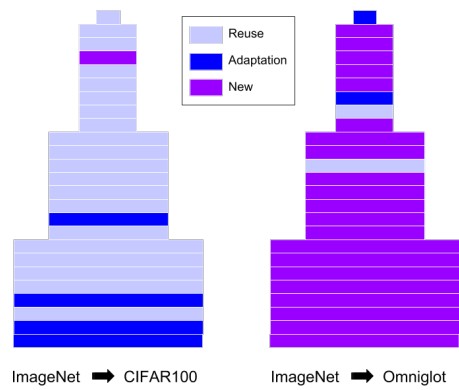

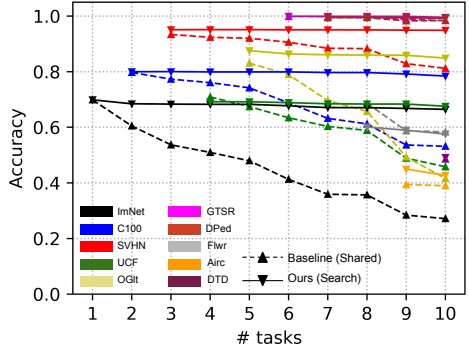

Figure 5: Visualization of searched architecture with learning two tasks sequentially. The search are based on the super model obtained by training ImageNet as first task. (a) and (b) shows searched architecture on CIFAR100 and Omniglot task respectively.

Figure 6: Comparison catastrophic forgetting effect between our proposed approach and baseline on visual decathlon dataset.

First, we run experiments on the permuted MNIST dataset. As mentioned in Section 3.2, we have two strategies when the search result gives "reuse" for a certain layer. We can either fix the reused weight during retraining without the risk forgetting, or we can apply tuning on the "reused" layers. As a baseline, we shown that simply updating all the layers with stochastic gradient descent (SGD) results in catastrophic forgetting (see Fig. 3 c). After training 10 tasks, the average accuracy dropped from 97.9% to 63.0%. With the elastic weight consolidation (EWC) (Kirkpatrick et al., 2017), the forgetting was alleviated to 96.8% average accuracy over 10 tasks. For our approach, we found that tuning the "reuse" layers by using simple $\ell_2$ based regularization on previous task parameter (i.e. $\|\theta_i - \theta_j\|_2^2$, where $\theta_i$ is the parameter for current task and $\theta_j$ is the $j$th task parameter that selected to reuse) is safe enough in terms of forgetting and both strategies can keep the overall accuracy as high as training each task individually (see Fig. 3 c). Encouraged by the above result, we additionally run experiments by tuning the "reuse" layers *without using any regularization*. In other words, we do not add any regularization to the parameters to mitigate forgetting among different tasks. The results is shown in Fig.3 c. As can be observed, here we almost have the same behavior as compared to using the $\ell_2$ based regularization. This may suggests that the learned architecture actually make sense, and the reused parameter is close to optimal for specific tasks.

We would like to note that simple $\ell_2$ based regularization employed above is not capable of preventing forgetting in general. This is because using too strong regularization would leads to model parameters that are very close to the original task, which would then prevent the model from learning new tasks. Smaller regularization helps, however, as the number of tasks increase, the distance between the last model parameter and the very first could be significant, even though each successive distances are small. For baseline model we need to use relatively large

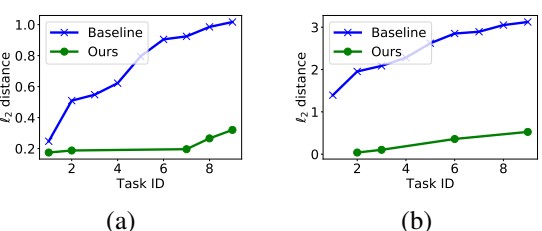

(a)                                   (b)

Figure 4: Distance between the tuned parameter at each task and the parameter of the very first task on VDD dataset. a) First layer parameter distance, and b) Last layer parameter distance. Baseline indicates the result from tuning all layers using SGD.

regularizations, and thus leads to worse performance of new tasks. Using smaller regularization in baseline leads to significant forgetting. Observing significantly less forgetting in the above experiment may suggests that the model parameters does not move much, or moved on the same level set of the loss surface.

Next we run a similar experiment on the visual decathlon dataset. All ten tasks are trained in order of ImageNet, CIFAR-100, SVHN, UCF101, Omniglot, GTSR, DPed, Flower, Aircraft and Textures. As a baseline we also train a model that shares all parts of the structure. Both models are trained with similar setting as in previous permuted MNIST experiments, except in our method we choose not to use any regularization this time due to the positive results we got from last experiment. As can be seen from figure 6, the performance from our approach with explicit learning of structure constantly out performs the baseline, suggests a lot less forgetting when transfer between different tasks.

To see if the 'reuse' layers are almost at an optimal position for current task to use, we perform an additional experiment. We calculate the $\ell_2$ distance between the original parameter and the tuned parameter after each tasks on the VDD dataset. Fig.4 shows $\ell_2$ distance between the parameters from the very first task to each of the tuned parameters in following tasks from the first and last layers. It is clear that the parameter does not move much from the original location as compared to the baseline, which explains why we observe less forgetting[2]. In addition, the moved distance from our methods is more or less the same scale across all layers. This may also attribute to the fact that the selection of utilization of parameters (i.e. structure) is explicitly learned, and thus the selected ones are more compatible with current task. Therefore, less tuning is required for the new task and hence smaller distance.

Experiments in this section indicates that learning structure is important. With the right structure all the relevant parameters from previous tasks can be used. Additionally, since the way to utilize these parameters are learned through structure learning, much less tuning is required for better performance on the new task, and forgetting can thus be minimized.

### 4.3 COMPARISON WITH OTHER METHODS

In this section, we compare our methods with other recent continuous learning methods – Lee et al. (2017b, DEN), Serrà et al. (2018, HAT), Kirkpatrick et al. (2017, EWC), Lee et al. (2017b, IMM), Rusu et al. (2016, ProgressiveNet), Fernando et al. (2017, PathNet), Nguyen et al. (2018, VCL). We compare the performance of various methods on the permuted MNIST dataset with ten different permutations. Since our model adds more parameters, for fair comparison we also train other methods with as many or more parameters as compared to our model. In particular, since our model tends to add new parameters at the first layer, for all methods we enlarge the first layer hidden size by ten times, so that theoretically they could learn exactly the same structure as our model. We ensure that all methods use similar amount of parameters, and the parameter usage of our methods does not exceed any of the other methods that we compare. We also tried to compare with Shin et al. (2017), however, we are unable to get reasonable performance, and thus, the result is not included. More details regarding this experiment can be found in appendix. The results are shown in Fig. 7a. It is clear that our method (either tuned with or without regularization) performs competitive or better than other methods on this task. This result suggests that although theoretically, structure can be learned along with parameter, in practice, the current optimization have a hard time achieving this. This in turn indicates the importance of explicit taking structure learning into account when learning tasks continuously.

Although both DEN and our method dynamically expand the network, our performance is much better, which is attributed to the ability of learning the new structure for different tasks. Additionally, our model performs competitive or better as compared to methods that completely avoids forgetting by fixing the learned weights, such as ProgressNet and PathNet, without having such restrictions.

Additionally, we performed experiments on split CIFAR-100 dataset (Lopez-Paz et al., 2017), where we randomly partition the classes of CIFAR-100 into 10 disjoint sets, and regard learning each of the 10-class classification as one task. Different from permuted MNIST, the split CIFAR-100 presents a sequential task where the input distribution is similar whereas the output distribution is different. We choose to use Alexnet as the network structure, and all methods are constrained to use similar number of parameters. This network structure contains three convolution and max pooling layers and two fully connected layers before the last classification layer. Comparative results with other

---

[2]Similar trend regarding the distance between parameters from tasks was found for all layers. In general, the higher up the layer the more the parameter moves for the baseline, whereas for our method the moved distance is typically very small

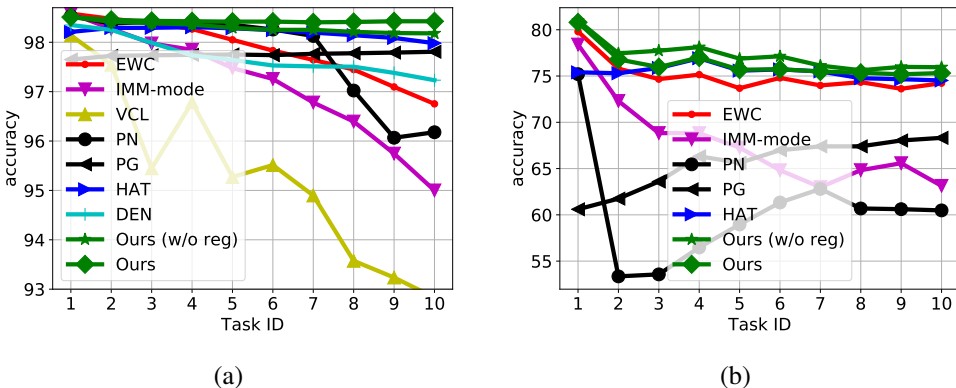

Figure 7: Comparative performance on a) permuted MNIST and b) split CIFAR-100 dataset. Methods include Kirkpatrick et al. (2017, EWC), Lee et al. (2017b, IMM), Fernando et al. (2017, PathNet (PN)), Rusu et al. (2016, Progressive Net (PG)), Serrà et al. (2018, HAT), Lee et al. (2017b, DEN), Nguyen et al. (2018, VCL), ours (w/o reg) denotes the case where finetuning for current tasks is done without using any regularization to prevent forgetting, and ous represents the case where the $\ell_2$ regularization is used.

methods is shown in Fig 7b. Similar trend as the MNIST experiment is observed in this experiment. Interestingly, for all tasks, our method always find structure that use new parameters for the last convolution layer and reuse the rest of the network parameters. It make sense that the lower layer features get shared, and the higher ones needs to be specific for different tasks, since the input data distribution has a lot of commonality. The fully connected layers are all selected to be "reused" instead of "new", and this may because of the relatively large capacity that is already flexible enough to fit the latter tasks.

## 5    RELATED WORK

Continual learning (Thrun & Mitchell, 1995) is a challenging problem, as models have the tendency to forget previously learned knowledge when learning on new information (Thrun & Mitchell, 1995; McCloskey & Cohen, 1989). This is referred as catastrophic forgetting problem in the literature. Early attempts to alleviate catastrophic forgetting often consists of memory system that store previous data and replay the sampled old examples with the new data (Robins, 1995), and similar approaches are still used today (Rebuffi et al., 2017b; Li et al., 2018; Lopez-Paz et al., 2017). Shin et al. (2017) learns a generative model from to capture the data distribution of previous tasks, and both generated samples and real samples from the current task is used to train the new model so that the forgetting can be alleviated for continual learning.

Another class of common method for mitigating catastrophic forgetting is through regularization which imposes constraints on the update of neural weights. Kirkpatrick et al. (2017) proposed elastic weight consolidation (EWC), which tries to minimize the change of weights that are important to previous tasks through the use of a quadratic constraint. Zenke et al. (2017) proposed to alleviate catastrophic forgetting by allowing individual synapse to estimate their importance for solving a learned task, then penalize the change on the important weights. Schwarz et al. (2018) divided the learning to two phases – progress and compress. During progress phase, the model make use of the previous model for learning the new task. In compression phase, the newly learned model is distilled into the old model by using EWC to alleviate forgetting. Serrà et al. (2018) proposed method that use attention mechanism to preserve previous' tasks performance. One could also completely avoid forgetting by preventing changes to previous task weights (see for example Rusu et al., 2016; Mallya & Lazebnik, 2018; Fernando et al., 2017).

Another class of methods for continuous learning is allowing the model to expand. Dynamically expandable networks (Lee et al., 2017a) select whether to expand or duplicate based on certain criteria on the new task. However, the model on the new task is forced to use the old structure from previous tasks. Similarly for progressive networks (Rusu et al., 2016). Our framework is more

flexible, as it allows model to choose whether to use previous structures. Pathnet (Fernando et al., 2017) select paths between predefined modules, and tuning is allowed only when an unused module is selected. Our method has more granularity, and does not have any restriction on tuning parameters from previous tasks.

Our work also relates to neural architecture search (Stanley & Miikkulainen, 2002; Zoph & Le, 2016; Baker et al., 2017; Liu et al., 2018), as we employ search methods for the implementation of structure optimization. In particular, DARTS (Liu et al., 2018), where a continuous relaxation for architecture search is proposed. This allowed more efficient structure optimization, and hence employed in our work.

## 6 CONCLUSION

In this work, we propose to explicit take into account structure optimization in a continuous learning setting. Empirically, we found that explicit learning of structure does lead to learning of sensible structures for each of the tasks. In particular, lower level features are shared more in case the input representations are more similar, and higher level representations tends to share when the high level semantic has commonality. Additionally, the catastrophic forgetting problem gets alleviated with structure optimization. We achieved reasonable performance on all tasks, and simultaneously minimized forgetting by only use simple $\ell_2$ based regularization. Further, we show that forgetting is small even without using any regularization on the selected reuse parameters. This suggests that structure learning plays a significant role in the catastrophic forgetting problem. Moreover, we achieved competitive or better performance as compared to all other methods on the permuted MNIST and split CIFAR-100 datasets.

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

# Appendices

## A  ADDITIONAL EXPERIMENTAL DETAILS FOR PERMUTED MNIST

For all MNIST experiment, we use fully connected layer with three hidden layer, each with 300 hidden units, and one shared output layer for our method. For all other methods except pathnet and progressive net we used 3000 units in the first layer and 300 for the rest. For pathnet, each module in the first layer has 300 units, and the result layers has 30 units. We use 16 modules per layer, and 5 layers for pathnet, and restrict each mutation to only use 3 modules besides the output layer. For progressive net, the first layer has 300 units for each task, and the rest layers each has 30 units. Therefore, all competitive methods are having more or the same number of parameters as our methods.

For variational continual learning (VCL Nguyen et al., 2018), we used the official implementation at `https://github.com/nvcuong/variational-continual-learning`. For fair comparison with other methods, we set the coreset size to zero for VCL.

For (Shin et al., 2017) we used implementation from `https://github.com/kuc2477/pytorch-deep-generative-replay`. We tried various hyper-parameter settings, however, we are unable to get reasonable results on permutated MNIST. Performance was reasonable when the number of tasks is within five (average performance at around 96%). When number of tasks go beyond five, performance drops on previous tasks is quite significant. Reaching 60%

For DEN we use the official implementation at `https://github.com/jaehong-yoon93/DEN`, and we used Serrà et al. (2018) implementation of HAT, EWC, and IMM at `https://github.com/joansj/hat`. We used our own implemention for for Progressive Network and PathNet. All methods are trained using the same permutations and same subset of training data.

## B  ADDITIONAL EXPERIMENTAL DETAILS FOR SPLIT CIFAR-100

For all CIFAR-100 experiment, we use an Alexnet like structure. It contains three convolution and max pooling layers followed by two fully connected layers. The convolution layers are of size (4,4), (3,3) and (2,2) with 64, 128 and 256 filters, respectively. All convolution layers are followed by max pooling layer of size (2,2) and rectified linear activations. The two fully connected layers each have 2048 hidden units.

## C  ADDITIONAL EXPERIMENTS ON VISUAL DECATHLON DATASET

In the multi-task continual learning experiments, the 10 tasks was trained in a random sequence except the first task was fixed to be ImageNet. This is just for fair comparison with other works such as Rebuffi et al. (2017a) and Mallya & Lazebnik (2018), they are all using a light weight module to adapt ImageNet pretrained model to other of the 9 tasks. In real case, the tasks can come in any order, thus our framework would be much more flexible. As the tasks are trained in sequence, a super model is maintained that all the newly created weights and task-specific layers are stored. In this ResNet-26 model, all the Batch Normalization (BN) layers are treated as task-specific, which means each task has its own sets of BNs. Here, we fixed the weight during retraining when "reuse" is selected in the search phase. This means that the results of previous tasks would not be affected, i.e. no forgetting. We leave the evaluation of forgetting in the context of VDD dataset as future work.

In Table 1, we compare the results using our approach with other baselines. "Individual" means that each task is trained individually and weights are initialized randomly. "Classifier" means that only the last layer classifier could be tuned while the former 25 layers are transfer from ImageNet pretrained model and kept fixed during training. In this case, each task only adds a task-specific classifier and BNs, thus the overall model size is small. "Adapter" add a 1x1 conv layer besides each 3x3 conv layer, and the outputs will be added before proceed to the next layer. Due to the lightweight 1x1 conv layer, each task will add approximately 1/9 of the whole model size. As shown in table 1, the results achieved by our framework is better than other baselines and the total

model size is similar to "Adapter" case. We can see that our approach gives best results in five out of nine tasks. Especially in task with small data size, e.g. VGG-Flowers and Aircraft, our method outperforms other baselines by a large margin.

Due to each choice has different parameter cost, we add a parameter loss function to $L_{val}$ to penalize the choices that cost additional parameters. And the value of the loss function is proportional to the product of the additional parameter size and its corresponding weight value $\alpha_c^l$. In table 2, we test it with three different scaling factor $\beta$ of the parameter loss. We found that the scaling factor $\beta$ can control the additional parameter size for each task. And we find that $\beta = 0.1$ gives the best average accuracy and can control the total model size approximate $2.3\times$ compared with the original model size.

| Model | ImNet | C100 | SVHN | UCF | OGlt | GTSR | DPed | Flwr | Airc. | DTD | avg. | #params |
|---|---|---|---|---|---|---|---|---|---|---|---|---|
| Individual | 69.84 | 73.96 | 95.22 | 69.94 | 86.05 | **99.97** | **99.86** | 41.86 | 50.41 | 29.88 | 71.70 | 58.96 |
| Classifier | 69.84 | 77.07 | 93.12 | 62.37 | 79.93 | 99.68 | 98.92 | 65.88 | 36.41 | 48.20 | 73.14 | 6.68 |
| Adapter | 69.84 | **79.82** | 94.21 | 70.72 | 85.10 | 99.89 | 99.58 | 60.29 | 50.11 | **50.60** | 76.02 | 12.50 |
| Search (Ours) | 69.84 | 79.59 | **95.28** | **72.03** | **86.60** | 99.72 | 99.52 | **71.27** | **53.01** | 49.89 | **77.68** | 14.46 |

Table 1: Results of the (top-1) validation classification accuracy (%) on Visual Domain Decathlon dataset. The total model size ("#params") is the total parameter size (in Million) after training the 10 tasks.

| | | ImNet | C100 | SVHN | UCF | OGlt | GTSR | DPed | Flwr | Airc. | DTD | Tot. |
|---|---|---|---|---|---|---|---|---|---|---|---|---|
| $\beta = 0.01$ | acc | 69.84 | 78.50 | 95.33 | 72.50 | 86.41 | 99.97 | 99.76 | 66.01 | 51.37 | 50.05 | 76.97 |
| | #params | 6.07 | 0.15 | 2.74 | 2.28 | 6.17 | 3.59 | 1.02 | 0.19 | 4.15 | 0.13 | 26.49 |
| $\beta = 0.1$ | acc | 69.84 | 79.59 | 95.28 | 72.03 | 86.60 | 99.72 | 99.52 | 71.27 | 53.01 | 49.89 | 77.68 |
| | #params | 6.07 | 0.34 | 1.19 | 1.32 | 3.19 | 0.02 | 0.27 | 0.16 | 1.86 | 0.04 | 14.46 |
| $\beta = 1.0$ | acc | 69.84 | 78.00 | 93.40 | 63.83 | 84.30 | 99.78 | 99.01 | 65.77 | 39.27 | 48.77 | 74.20 |
| | # params | 6.07 | 0.04 | 0.03 | 0.12 | 0.66 | 0.02 | 0.01 | 0.02 | 0.35 | 0.02 | 7.34 |

Table 2: Comparison of (top-1) validation classification accuracy (%) and total model size (in Million) on Visual Domain Decathlon dataset with parameter loss factor $\beta$ of 0.01, 0.1, 1.0.

