# OpenReview forum: "Continual Learning via Explicit Structure Learning"
_ICLR.cc/2019/Conference_

### Official Review · AnonReviewer1 · 2018-11-01
**Interesting idea, but needs a stronger experimental justification**

**Rating:** 4
**Confidence:** 4

**Review:**

The proposed approach aims to mitigate catastrophic forgetting in continual learning (CL) problems by structure learning: determining whether to reuse or adapt existing parameters, or initialise new ones, when faced with a new task. This is framed as an architecture search problem, applying ideas from Differentiable Architecture Search (DARTS). The approach is verified on the Permuted MNIST dataset and evaluated on the Visual Decathlon, showing an improvement.

I think this is an interesting idea with potential, and is worth exploring, and the paper is well-structured and easy to follow.

Unfortunately, I feel the paper fails to consider recent work on CL, both in terms of discussion and benchmarking. The only previous work that is compared is EWC, on permuted MNIST, and the Visual Decathlon performance is only compared to simple baselines (such as adding an adapter or fine tuning) which makes it difficult to gauge the contribution.
There are recent works, some with better results on more difficult problems, such as Variational Continual Learning [1], Progress and Compress [2], or (Variational) Generative Experience Replay [3][4].
Given the approach is based on dynamically adding parameters or modules, Progressive Networks and Dynamically Expandable Networks (both cited) are especially relevant and should be compared (I believe the former may be related to the “adapter” baseline, but this should be made explicit).

I have some questions / discussion points:
- What's the intuition behind implementing the “adapt” operator as additive bias over the previous weights, rather than just copying the previous weights and fine tuning?
- In the general case, if the architecture search is a continuous relaxation (softmax combination of operators), why is the "adapt" operator necessary? Wouldn't this already be a linear combination of new and old parameters? (In the example case of a 1×1 adaptor it makes sense, but this is a special restricted case which adapts with a smaller set of parameters)
- How is the structure regulariser backpropagated into the parameters of each layer? As I understand, it is composed of a constant discrete term z (number of parameters in each option), multiplied by architecture softmaxes alpha; the gradient with respect to each alpha is a constant, and so this has the effect of scaling the gradients of each operator.
- For the "reuse - tuned" case, isn’t the model effectively maintaining a new network for each task?

I also have a number of other comments:
- Reference to figure in page 6 should be figure 4, not 5.
- I think the readability of the paper would benefit from another few proofreads; there are a number of grammatical issues throughout, and several sentence fragments, eg. in the top para of page 2: “..., it has the potential to encourage information sharing. Since now the irrelevant part can be handled…”.

I would encourage the authors to strengthen the experimental comparison by incorporating stronger, external baselines, and improving some of the minor writing issues.

[1] Nguyen, Cuong V., et al. "Variational Continual Learning." ICLR, 2018.
[2] Schwarz, Jonathan, et al. "Progress & Compress: A scalable framework for continual learning." ICML, 2018.
[3] Shin, Hanul, et al. "Continual learning with deep generative replay." NIPS, 2017.
[4] Farquhar, Sebastian, and Yarin Gal. "Towards Robust Evaluations of Continual Learning." arXiv, 2018.

---

> ### Author Response · Authors · 2018-11-17
> **Response**
>
> Thanks for your feedback. We have added additional results comparing our method with other more recent and relevant methods. Please refer to the updated manuscript.
>
> Regarding the questions:
> - What's the intuition behind implementing the “adapt” operator as additive bias over the previous weights, rather than just copying the previous weights and fine tuning?
>
> The role of adaptor is to strike a balance between number of parameters and performance. As mentioned in the end of section 3.1, we have different cost for select each option. Adaptor provides a way of using and modifying previous representation without incurring any forgetting by adding a relatively small amount of parameter overhead.
>
> - In the general case, if the architecture search is a continuous relaxation (softmax combination of operators), why is the "adapt" operator necessary? Wouldn't this already be a linear combination of new and old parameters? (In the example case of a 1×1 adaptor it makes sense, but this is a special restricted case which adapts with a smaller set of parameters)
>
> In the adaptor case, when searching the combination of the old parameters with 1x1 conv forms an option. For example, in case we have two options, reuse and adaptor, the softweight is over the original parameter and the original parameter plus adaptor combined, so here the second part is treated as one option. To some extend what you are suggesting is true, however, this does not exactly corresponds to what is happening (as we explained above).
>
> - How is the structure regulariser backpropagated into the parameters of each layer? As I understand, it is composed of a constant discrete term z (number of parameters in each option), multiplied by architecture softmaxes alpha; the gradient with respect to each alpha is a constant, and so this has the effect of scaling the gradients of each operator.
>
> In our implementation, the structure regularizer does not backprop to the parameters of each layer. Instead, the regularizer serves as a penalty for different choices, and thus has effect on the magnitude of alphas. Since alpha controls the weight for different options, this would influence the choice of different options during structure learning.
>
> - For the "reuse - tuned" case, isn’t the model effectively maintaining a new network for each task?
>
> No. When the model is reused, the parameters are tuned, and the tuned parameter is used both for current tasks that it is finetuned on as well as all previous tasks.

---

> > ### Comment · AnonReviewer1 · 2018-11-24
> > **Paper has improved, but I believe it needs more work**
> >
> > I have read the authors' response and the updated manuscript, and I applaud their efforts to improve the paper.
> >
> > However, while the paper is improved, regrettably, I still feel it falls short of publication.
> >
> > In particular, as I stated in my original review, the comparisons (even with the newly added baselines) are only applied to permuted MNIST, and the VDD performance baselines are quite simple. Permuted MNIST, while used in the past, is arguably no longer considered a strong evaluation, as the tasks are relatively independent [4].
> > The references I suggested don't appear in the paper, except for [4], which is only mentioned in passing to introduce single-headed MNIST.
> >
> > Finally, the writing issues still seem to persist, such as sentence fragments and a few typos throughout.
> > eg. in the newly added section 4.3: "In particular, since our model tends to add new parameters at the first layer. For all methods..."
> >
> > As with my original review, I think this approach has potential, but I think the writing issues need to be addressed, references added, and comparisons performed on VDD or another strong dataset.

---

> > > ### Author Response · Authors · 2018-11-27
> > > **Response**
> > >
> > > We have added variational continual learning result. The result was not added in the first version because running the VCL with more parameters uses a lot of memory, and thus can only run on CPU, which is a bit slow.
> > >
> > > We tried deep generative replay, however we are not able to get reasonable results on permuted MNIST with 10 permutations. We tried various hyper-parameter settings, and performance was reasonable when the number of tasks is within five (average performance at around 96%). When number of tasks go beyond five, performance drops on previous tasks is quite significant, some tasks dropped to ~60%
> > >
> > > We have added suggested references in related work.`

---

### Official Review · AnonReviewer3 · 2018-11-02

**Rating:** 4
**Confidence:** 5

**Review:**


This paper proposes a new approach to mitigate the catastrophic forgetting for continual learning. The model is composed to the neural architecture search and parameter learning based on the intuition that largely different tasks should allow to use different network structure to train them. In structure learning, they introduce three candidate to decide network architecture, reuse, adaptation and new. In the experiments, they show that their model outperforms SGD and EWC.

Basically, the intuition of structure learning and the validation of that is straight forward and easy to follow. However, I’m not sure that the proposed model can outperform the recent continual learning methods, such as IMM(Lee et al, 2017), DEN or  RCL(Ju Xu et al, 2018). There is only a relatively weak(and old) comparison with l2, and EWC.

-	In the equation (4), I wonder that, in the model, the hyperparameter(lambda_i or beta_i) of regularizer looks different according to the task, is it correct?
-	As shown in the Fig. 2) three choice-reuse, adaptation, and, new, is decided in the layer level. But with a semantic intuition, such that two different task can share specific features and simultaneously each of them requires the different neural space to learn discriminative ones at layer l, it seems better if the model could search structure much flexible. Is there some of experimental trial or plan about these kind of joint-adoption?
-	What is the main contribution of adaptation? I wonder that only reuse and new can work well including the role of adaptation, or not.
-	Is there any experiments to compare the recent continual learning methods(as I mentioned), in terms of AUC(or accuracy) and the network capacity?

Minor remarks,
Page 3: 	“is been” -> is
	“unlikely”-> unlike
Page 4: 	“sharealbe” -> shareable
Page 5: 	“, After” -> , after
	“permuated” -> permuted
Page 6:	“Fig. 5” -> Fig. 4

---

> ### Author Response · Authors · 2018-11-17
> **Response**
>
> Thanks for your feedback. We have added additional results comparing our method with other more recent and relevant methods. Please refer to the updated manuscript.
>
> Regarding the questions:
> -	In the equation (4), I wonder that, in the model, the hyperparameter(lambda_i or beta_i) of regularizer looks different according to the task, is it correct?
>
> Yes they can be different for each tasks, this is more of a design choice. However, in our implementation and experiments, to make things easier, we just used the same hyperparameter for all tasks.
>
> -	As shown in the Fig. 2) three choice-reuse, adaptation, and, new, is decided in the layer level. But with a semantic intuition, such that two different task can share specific features and simultaneously each of them requires the different neural space to learn discriminative ones at layer l, it seems better if the model could search structure much flexible. Is there some of experimental trial or plan about these kind of joint-adoption?
>
> This is a very good point. Ideally we would like to be able to do more finer grained search, and that is definitely desired. In practice, we could only make the search space more restricted so that the search can be done in a more efficient manner. Of course one is not restricted to use only the options that we provided in our implementation. More finer grained and search is definitely possible, for example, learning to share at filter/neuron level instead of layer level. This is more of a balance between training efficiency and final performance. The current implementation highlights the importance of taking structure into account. However, one should not limit themselves with only the options that we demonstrated. As long as the search space is reasonably sized and operations are plausible, it could be incorporated in our framework. This leads to interesting future work directions.
>
> -	What is the main contribution of adaptation? I wonder that only reuse and new can work well including the role of adaptation, or not.
>
> The role of adaptation is to strike a balance between number of parameters and performance. As mentioned in the end of section 3.1, we have different cost for select each option. Adaptor provides a way of using and modifying previous representation without incurring any forgetting by adding a relatively small amount of parameter overhead.

---

### Official Review · AnonReviewer2 · 2018-11-03
**Review of "Continual Learning via Explicit Structure Learning"**

**Rating:** 4
**Confidence:** 4

**Review:**

The paper considers the problem of sequential learning where data access for the previous tasks is completely prohibited. Authors propose a conceptually simple framework to learn structures (it is the selection of reusing, adapting previously learned layers or training new layers) as well as corresponding parameters in the sequential learning.

The paper is potentially interesting and providing possibly important framework for life-long learning. It is well written in most of cases and easy to follow (however I got the impression that the paper was rushed in the last minute; there are some trivial typos and very low resolution images etc.)

However, I have a huge concern about the empirical evaluations.  This area is really huge and has attracted lots of interest from many researchers, meaning that we lots of methods to compare. Nevertheless, authors only focus on providing insights on effects of different components of the propose model. This is also critical but comparing against state-of-the-arts is also very important. Especially, comparing against Lee et al 2017 seems essential. I can see the difference against that paper from the authors' argument in the related work, but that is the difference not comparison. It would be great to compare the performances as well as the number of increased memory sizes as the number of task increases.

Moreover, the details should be provided; for instance provide the explicit form of R(s).

---------------------------------------------

Thanks for the update. But are they fair comparisons (evaluation only in terms of accuracy)? Different methods expand the network different amount. Hence, they should be compared on this metric too.

---

> ### Author Response · Authors · 2018-11-17
> **Response**
>
> Thanks for your feedback. We have added additional results comparing our method with other more recent and relevant methods. Please refer to the updated manuscript.

---

> ### Author Response · Authors · 2018-11-28
> **Response after update**
>
> Are they fair comparisons (evaluation only in terms of accuracy)? Different methods expand the network different amount. Hence, they should be compared on this metric too.
>
> As mentioned in the paper, we make sure that all methods use similar amount of parameters. In particular, we make sure that all other methods at least match the number of parameters for our final model (after 10 tasks). In other words, all compared methods has same or more capacity as compared to our model, and we believe this comparison represents a fair comparison.
>
> We agree that the expansion amount is also an important metric, and we will this metric in the final version.

---

### Author Response · Authors · 2018-11-17
**Paper revision summary**

We thank all reviewers for providing constructive feedback that further improves the paper. We have highlighted the changes in text by using blue color. Minor editing changes are not marked. In this update revision we did following changes.

1) We added more analysis on forgetting, which we think provides more insights into the method. In addition to use simple L-2 based regularization we finetuned our model without using any regularization, and we still obtained interesting result where the forgetting is minimal. This further suggests the importance of structure learning when learning continual tasks.

2) As all reviewers suggested, we added more comparisons to more recent, existing methods. In particular, we compared ours with the more recent methods such as  dynamically expandable network, incremental moment matching, progressive network, hard attention to task, etc on permuted MNIST dataset. We show that our method is performs competitive or better as compared to all these method.

3) Provided more details in appendix

4) Corrected editorial errors as pointed out by reviewers.

Due to the time limit, we only completed experiments on permuted MNIST. Additionally we are also running experiment on split MNIST so that we have more comparisons, and we will update another version with those results before the deadline.

---

> ### Author Response · Authors · 2018-11-27
> **2nd paper revision**
>
> We have added one more experiment on split CIFAR-100. As reviewers suggested, MNIST dataset may not be a strong evaluation set, and therefore we added CIFAR-100 experiments since it represent a more realistic settings.

---

### Meta-Review · Area_Chair1 · 2018-12-13

**Confidence:** 5
**Recommendation:** Reject

**Metareview:**

The paper presents a promising approach for continual learning with no access to data from the previous tasks. For learning the current task, the authors propose to find an optimal structure of the neural network model first (select either to reuse, adapt previously learned layers or to train new layers) and then to learn its parameters.

While acknowledging the originality of the method and the importance of the problem that it tries to address, all reviewers and AC agreed that they would like to see more intensive empirical evaluations and comparisons to state-of-the-art models for continual learning using more datasets and in-depth analysis of the results – see details comments of all reviewers before and after rebuttal.
The authors have tried to address some of these concerns during rebuttal, but an in-depth analysis of the results (evaluation in terms on accuracy, efficiency, memory demand) using different datasets still remains a critical issue.

Two other requests to further strengthen the manuscript:
1) an ablation study on the three choices for structural learning (R3), and especially the importance of ‘adaptation’ (R3 and R1)
The authors have tried to address this verbally in their responses but a proper ablation study would be desirable to strengthen the evaluation.
2) Readability and proofreading of the manuscript is still unsatisfying after revision.